# Modified Sol–Gel Synthesis of Mesoporous Borate Bioactive Glasses for Potential Use in Wound Healing

**DOI:** 10.3390/bioengineering9090442

**Published:** 2022-09-05

**Authors:** Farzad Kermani, Hossein Sadidi, Ali Ahmadabadi, Seyed Javad Hoseini, Seyed Hasan Tavousi, Alireza Rezapanah, Simin Nazarnezhad, Seyede Atefe Hosseini, Sahar Mollazadeh, Saeid Kargozar

**Affiliations:** 1Department of Materials Engineering, Faculty of Engineering, Ferdowsi University of Mashhad (FUM), Mashhad 9177948564, Iran; 2Thoracic Surgery Department, Ghaem Hospital, Mashhad University of Medical Sciences, Mashhad 917699311, Iran; 3Department of General Surgery, School of Medicine, Mashhad University of Medical Sciences, Mashhad 9177948564, Iran; 4Surgical Oncology Research Center, Imam Reza Hospital, Mashhad University of Medical Sciences, Mashhad 9176999311, Iran; 5Department of Medical Biotechnology and Nanotechnology, Faculty of Medicine, Mashhad University of Medical Sciences, Mashhad 9177948564, Iran; 6Tissue Engineering Research Group (TERG), Department of Anatomy and Cell Biology, School of Medicine, Mashhad University of Medical Sciences, Mashhad 9177948564, Iran

**Keywords:** mesoporous bioactive glass (MBGs), sol–gel synthesis, borate 1393B3 glass, antibacterial activity, tissue engineering

## Abstract

In this study, we successfully utilized nitrate precursors for the synthesis of silver (Ag)-doped borate-based mesoporous bioactive glass (MBGs) based on the 1393B3 glass formulation in the presence of a polymeric substrate (polyvinyl alcohol (PVA)) as a stabilizer of boric acid. The X-ray diffraction (XRD) analysis confirmed the glassy state of all the MBGs. The incorporation of 7.5 mol% Ag into the glass composition led to a decrease in the glass transition temperature (T_g_). Improvements in the particle size, zeta potential, surface roughness, and surface area values were observed in the Ag-doped MBGs. The MBGs (1 mg/mL) had no adverse effect on the viability of fibroblasts. In addition, Ag-doped MBGs exhibited potent antibacterial activity against gram-positive and gram-negative species. In summary, a modified sol–gel method was confirmed for producing the Ag-doped 1393B3 glasses, and the primary in vitro outcomes hold promise for conducting in vivo studies for managing burns.

## 1. Introduction

Chronic non-healing wounds (CNHW) constitute a significant health concern throughout the world and affect the life quality of nearly 2.5% of United States people [1]. The financial burden is higher than the $25 billion/year spent on CNHW treatment in just the United States [1]. Therefore, numerous attempts have been made to manage CNHW, including the use of traditional wound dressings (e.g., gauze and ointments) and advanced dressings (e.g., biodegradable and biocompatible materials) [2,3]. However, bacterial infections are still among the most challenging issues ahead of treating CNHW in the clinic and may impede wound healing. In this subject, eradicating multidrug-resistant (MDR) bacteria (e.g., *Staphylococcus aureus* and *Pseudomonas aeruginosa*) is an unsolved problem and needs innovative and novel therapeutic approaches. Accordingly, the use of antibacterial materials is of utmost importance for preventing bacteria contamination and colonization in the wound bed. Up to now, plentiful natural and synthetic antibacterial substances have been developed and used for killing or inhibiting bacteria; bioactive glasses (BGs) represent versatile materials with the ability to act against both gram-positive and gram-negative species, either in planktonic or sessile forms. BGs can negatively affect bacteria growth through changes in the environmental pH and increased osmotic pressure [4].

In 1969, Prof. Larry Hench invented the parent of biocompatible glasses (45S5 bioglass^®^) with the composition of 45 SiO_2_, 24.5 Na_2_O, 24.5 CaO, and 6 P_2_O_5_ (wt.%) for treating hard tissue damages and disorders, such as bone fractures [5]. These synthetic materials exhibit outstanding biological properties, including excellent biocompatibility, the ability to bond to hard and soft tissues, and promotion of angiogenesis [6,7,8]. Over the years, specific types and compositions of BGs have been produced and applied for managing disorders outside the skeletal system [9,10]. In this sense, borate-based BGs have been synthesized in diverse formulations for the treatment of soft tissue injuries, either alone or in combination with biopolymers [11,12,13]. For example, borate-based glass micro-fibers were previously reported to be useful in dermal repairing in vivo owing to their fast degradation and releasing ions (e.g., boron (B) or calcium (Ca)) into the surrounding area, which could support the migration of epidermal cells and regulate the wound healing process [14]. Besides, borate-based bioactive glass was suggested for wound healing, releasing growth factors and cytokines, increasing RNA synthesis, enhancing extracellular matrix turnover, and boron delivery in soft tissue engineering [15]. For instance, the biodegradable cotton-like fiber pad borate glass 13-93B3 was reported as a novel wound dressing for accelerating wound healing [16,17]. The mentioned fibers could contribute to the healing of long-term wounds in diabetic patients. Given the importance of managing bacterial infections, a series of metallic elements (e.g., silver (Ag)) were successfully incorporated into the basic composition of borate BGs for inhibiting biofilm formation and accelerating wound healing [18]. It has been well documented that Ag^+^ ions can enhance the production of reactive oxygen species (ROS) (e.g., OH*, O_2_*^−^, and H_2_O_2_) and thereby damage proteins, DNA, RNA, and lipids inside bacteria and eventually cause cell death [19,20]. 

In the present study, we could successfully synthesize Ag-doped 1393B3 borate mesoporous BGs (MBGs) using nitrate precursors by the sol–gel route in the presence of a polyvinyl alcohol (PVA) substrate. To the best of our knowledge, it is the first report that confirms the usability of the sol–gel method for the synthesis of mesoporous borate-based BGs using nitrate precursors, providing a great possibility for preparing inexpensive glass samples for tissue engineering applications.

## 2. Materials and Methods

### 2.1. Glass Synthesis

The Ag-doped 1393B3 borate MBGs were synthesized in a 54.6B_2_O_3_–(22.1-X) CaO–XAg_2_O–7.6MgO–7.9K_2_O–6.1Na_2_O–1.7P_2_O_5_ (X = 0, 1, 2.5, 5, 7.5) multi-component system. For this aim, appropriate amounts of reagents, including B(OH)_3_, Ca(NO_3_)_2_.4H_2_O, AgNO_3_, Mg(NO3)_2_.6H_2_O, KNO_3_, NaNO_3_, and (C_2_H_5_)_3_PO_4_ (Triethyl phosphate (TEP)) were defined using HSC chemistry^®^ software (HSC chemistry^®^ 9.4, Outotec, Espo, Finland) to obtain 10 g of the desired BGs (Table 1). In the experimental section, 10 g of B(OH)_3_ was dissolved in 200 mL absolute ethanol (Merck, Darmstadt, Germany) at 70 °C for 45 min by using a magnetic stirrer at a speed of 1000 rpm. The nitrate reagents were then dissolved in deionized water and added to the B(OH)_3_-containing solution in 45 min intervals. TEP was hydrolyzed in the presence of deionized water and then it was added to the obtained solutions (batch 1). In another batch, 10 g of PVA (M_W_ = 8800 g/mol) was dissolved in 300 mL of deionized water for 60 min at a temperature of 80 °C for 2 h (batch 2). The pH of batch 2 was increased up to 14 by using ammonium hydroxide solution (25% NH_3_ in H_2_O). After that, batch 1 was added to batch 2 with a drop rate of 30 drops/min under constant stirring, and the gel-like samples were immediately obtained. The aging process was carried out by storage of the prepared samples in sealed bottles for 7 days. In order to dry the samples, they were freeze-dried for 48 h. To initially remove the polymeric substrate, the samples were then pre-treated at a temperature of 250 °C for 24 h. Finally, the samples were heat-treated at 600 °C at a rate of 1 °C/min in the air. Figure 1 shows the schematic presentation of the process designed for synthesizing the sol–gel borate BGs.

### 2.2. Characterizations

#### 2.2.1. Thermal Behavior

In order to determine the thermal behavior of the glasses, the freeze-dried samples were analyzed using thermogravimetric analysis (TGA) and differential thermal analysis (DTA) analyses (STA 503, BAHR, Hullhorst, Germany) with a heating rate of 10 °C/min in the air. To discuss the glass transition temperature (T_g_) data, the Ag-doped borate-based BG compositions were adapted from SciGlass database version 7.12. The obtained data from the database were processed using SPSS Modeler (version 18.22, IBM, Armonk, NY, USA). It should be noted that we removed the duplicated data and replaced them with their average.

#### 2.2.2. Elemental Composition Analysis

The compositions of the calcined MBG particles were analyzed by inductively coupled plasma atomic emission spectroscopy (ICP-AES, Spectro Arcos, Kleve, Germany) after being digested in the aqua regia.

#### 2.2.3. XRD Analysis

The phase composition of the synthesized Ag-doped 1393B3 MBGs was investigated using X-ray diffraction (XRD) analysis (D8-Advance, Bruker, Karlsruhe, Germany) before and after immersion in SBF. The instrument conditions were set to do scanning at a 2θ range of 20–80°. The Cu-Kα radiation was employed with a step size of 0.05° and a time per step of 2 s. The characteristics of the crystalline HAp were studied using the Rietveld refinement technique (Profex 4.2.2 package, Solothurn, Switzerland) after the glass phase transformation in SBF. 

The rate constant (K) of the phase transformation of the MBGs to crystalline ceramic phase was calculated as suggested in Equation (1) [21].
K = −ln (Ca/C0)/T(1)
where C0 and Ca represent the initial amount of the crystalline ceramic after the first incubation time in SBF (i.e., 24 h (86,400 s)) and the amount of HAp after a time of T (s), respectively.

#### 2.2.4. FTIR Study

The primary bands of the synthesized MBGs, including B-O bands, were investigated using Fourier-transform infrared spectroscopy (FTIR) analysis on pellets consisting of a fixed amount of sample and KBr (Thermo Nicolet AVATAR 370, USA) over the range of 400–4000 cm^−1^. Furthermore, the main bands of the hydroxy carbonated layer (HCA), i.e., P-O, O-H, and C-O bands, were investigated in the SBF-incubated glasses.

#### 2.2.5. DLS and Surface Charge

The particle size values of the synthesized MBGs were measured by dynamic light scattering (DLS) (Vasco3, Cordouan, France) analysis. The zeta (ζ) potential and mobility values of the samples were measured using Zeta potential analyzer (NANO-flex^®^ II, Thermo Fisher Scientific, Waltham, MA, USA). For this purpose, the MBG powders (0.01 g) were first dispersed in absolute ethanol (10 mL) by applying ultrasonic waves (FR USC 22 LQ, 400 w, 20%, Taiwan) for 5 min and then introduced to the instrument.

#### 2.2.6. Electron Microscopy Observations

In order to observe the surface morphology, the samples were first gold sputter-coated and then introduced to field emission scanning electron microscopy (FESEM) (MIRA3, TESCAN, CZ) before and after incubation in SBF. The effect of the Ag-doped MBGs on the particle size was investigated by using transmission electron microscopy (TEM) (EM 208S, Philips, Amsterdam, The Netherlands). In addition, the impact of the dopant concentrations on the surface topography and roughness of the glasses was evaluated by AFM analysis (Nano Wizard II; JPK Instruments, Berlin, Germany). It is mentioned that TEM and AFM analyses were carried out through dispersion of 0.01 g of the MBGs powders in 30 mL of absolute ethanol with the assistance of ultrasonic waves (FR USC 22 LQ, 400 w, 20%, Taiwan) for 10 min. Then a drop of the suspensions was picked up with a lam and standard TEM grade for AFM and TEM analysis. 

#### 2.2.7. N_2_ Adsorption-Desorption Analysis

The mesoporous characteristics of the glass particles were determined using Brunauer-Emmett-Teller (BET) and Barrett–Joyner–Halenda (BJH) analyses (Quantachrome Instrument, Japan). Before the test, the glass powders were degassed at 250 °C for 6 h in a vacuum process.

### 2.3. Biological Evaluations

#### 2.3.1. In Vitro Bioactivity Assessment

According to Kokubo’s method [22], we prepared SBF to evaluate the bioactivity of the synthesized MBG particles. For this purpose, 0.15 g of the BG particles were incubated in 100 mL of SBF, as reported in [23]. Then the samples were shaken at a speed of 20 rpm at the temperature of 37 °C for 1, 3, and 7 days. Meanwhile, the pH changes of the samples were measured by using a digital pH meter (AZ pH Meter 86552, Taiwan). The phase, morphology, and the released ions concentration were detected using XRD, FTIR, FESEM, and ICP-AES analyses.

#### 2.3.2. Cell Viability

MTT ([3-(4,5-dimethyl-2-thiazolyl)-2,5-diphenyl tetrazolium bromide]) assay was performed to reveal the impact of the borate-based MBGs on the viability of mouse fibroblasts (NIH-3T3 cell line; National Cell Bank, Pasteur Institute of Iran). For this purpose, 1 × 10^4^ cells were plated in 96-well cell culture treated plates and cultured in RPMI-1640 supplemented by 10% fetal bovine serum (FBS) and 1% penicillin/streptomycin solution (Gibco, Waltham, MA, USA). The 1, 3, and 5 mg/mL of each sample were incubated with RPMI for 24 h in order to prepare a conditioned media. After 24 h, the cell culture media were replaced with the conditioned counterparts, and the cells were further cultured for an additional 24 h. Then the media were pulled out and replaced with the MTT solution (5 mg/mL). After 4 h, the MTT medium was aspirated and replaced with dimethyl sulfoxide (DMSO) (Sigma-Aldrich, Burlington, MA, USA) and shaken for 10 min. Finally, the optical density of the cell culture wells was read using a microplate reader (Synergy HT, BioTek, Winooski, VT, USA) at 570 nm.

#### 2.3.3. Antibacterial Activity

Gram-positive bacteria *Staphylococcus aureus* (*S. aureus*) (PTCC: 1112) and *Bacillus cereus* (*B. cereus*) (PTCC: 1247), as well as gram-negative bacteria *Escherichia coli* (*E. coli*) (PTCC: 1330) and *Pseudomonas aeruginosa* (*P. aeruginosa*) (PTCC: 1074), were cultured in the nutrient broth (NB) in an incubator at 37 °C for 24 h. The MBG particles were dissolved in a solvent containing deionized water and dimethyl sulfoxide (5 *v*/*v* DMSO) to obtain extracts of the BGs. The prepared samples were then incubated with bacteria for 24 h. Finally, the number of viable bacteria was counted, and the antibacterial activity of the glasses was calculated by using Equation (2), as follows:R = log(B/A) − log(C/A) = log(B/C)(2)
where R represents the antibacterial activity of the samples, A is the average number of viable bacteria before the test, B and C represent the average number of bacteria in the control group (DMSO) and experiment group (MBGs) after 24 h of incubation.

### 2.4. Statistical Analysis

The results obtained from ICP, particle size, ζ potential, cell viability, and antibacterial analyses were performed at least three times, and the data were represented as mean ± standard deviation (SD). The data was statistically analyzed via the one-way ANOVA analysis (GraphPad Prism, 8.4.3(686), New York, NY, USA) followed by post hoc analysis. (* *p* ≤ 0.05, ** *p* ≤ 0.01, *** *p* ≤ 0.001, **** *p* ≤ 0.0001).

## 3. Results

### 3.1. ICP Results

The ICP results of the MBGs digested in the aqua regia solution are represented in Table 2. The data confirmed that all of the compositions were almost matched with the designed compositions. 

### 3.2. DTA/TGA Analysis

The DTA and TGA graphs of all samples after performing the freeze-drying are illustrated in Figure 2A–E. In general, the endothermic peaks at temperatures lower than 100 °C are related to physically adsorbed water in PVA and other compound structures [24]. The endothermic peaks around 230–250 °C are associated with the elimination of water from the PVA structure and the polyene formation [24]. The peaks around 400–500 °C can be attributed to intramolecular condensation of decomposition of polyene to acetaldehyde, benzaldehyde, acrolein, cis, and trans derivatives. The peaks around 500–600 °C belong to the oxidation of the remained structural groups of PVA [24]. Moreover, the endothermic peaks around between 500 and 700 °C could be related to the decomposition of the remaining nitrate groups [23]. According to the DTA graphs, the glass transition of the Ag-free MBGs, 1, 2.5, 5, and 7.5 Ag-doped BGs were about 573, 540, 532, 529, and 526 °C. As can be seen in the TGA graphs, the major weight loss of the samples is related to the thermal decomposition of the PVA structure. Increasing the weight in the TGA graphs at a temperature higher than 550 °C can be associated with the crystallization of oxide phases [25]. The decision tree was considered based on the composition of this study, with a focus on borate-based bioactive compounds containing Ag or Ag_2_O (Figure 3A). The frequency distribution histogram of B_2_O_3_, Ag_2_O, and T_g_ in the extracted data is shown in Figure 3B. According to the data, doping of Ag or Ag_2_O to borate-based compounds reduces their T_g_. Based on the data in Figure 3C, the amount of T_g_ (y) as the function of the concentration of Ag_2_O (x) is y = −34.48 ln(x) + 463.54.

### 3.3. DLS Characterization

The particle size (Dv 50 and Dv 90) values of the prepared borate MBGs are represented in Table 3. The measured Dv 90 values for the Ag-free, 1, 2.5, 5, and 7.5 mol% Ag-doped samples were 105 ± 8, 50 ± 10, 40 ± 6, 25 ± 5, and 54 ± 9 nm, respectively. Compared with the Ag-free sample, a significant decrease and an increase (*p* < 0.05) were observed for the particle size values of Ag-doped samples, respectively.

The ζ potential values of the synthesized MBGs are shown in Table 3. The measured zeta potential values showed a significant increase (*p* < 0.05) from −12 ± 2 mV to −16 ± 2, −18 ± 3, −20 ± 2, and −22 ± 1 mV for the Ag-free glasses in comparison with their 1, 2.5, 5, and 7.5 mol% Ag-doped counterparts. The same trend was observed in the mobility of the mentioned samples; the values ranged from −0.89 ± 0.2 μ/s/V/cm for the Ag-free MBGs to −1.07 ± 0.1, −1.18 ± 0.1, −1.28 ± 0.1, −1.58 ± 0.3 μ/s/V/cm for 1, 2.5, 5, and 7.5 mol% Ag-doped samples, respectively.

### 3.4. XRD Patterns

The results of XRD, along with the Rietveld analyses before and after the MBGs immersion in SBF, are shown in Figure 4A–E. Besides, the results of the calculation of crystallinity, crystallite size, lattice constant, and rate constant of the BGs phase transformation to HAp are reported in Table 4. The XRD patterns of the BGs before immersion in SBF confirmed a major amorphous state of the synthesized particles with traces of crystalline calcium borate (CaB_2_O_4_, ICDD ref. cod. 01-076-0747, orthorhombic) (Figure 4A–D). Substitution of Ca with 1, 2.5, 5, and 7.5% Ag decreases the amounts of crystallized calcium borate from 11% to 7, 4, 4, and lower than 1 wt.%, respectively. The XRD graphs clearly indicate the presence of crystallized HAp phase (ICCDD ref. cod. 9-0432) in all the SBF-immersed glasses with a crystallinity degree lower than 40% and a crystallite size lower than 43 nm after 7-days of immersing.

After 7-days of immersion in SBF, the crystallinity degree values of HAp in the Ag-free, 1, 2.5, 5, and 7.5 mol% Ag-doped MBGs were 35, 41, 39, 36, and 20%, respectively. The crystallite size values of the mentioned samples were 38, 43, 41, 35, and 31 nm, respectively. According to the data presented in Table 4, the lattice constants (a = b, c) of the formed HAp showed an increase in the MBGs immersed in SBF for 7 days compared to the immersed samples for 3 days. In addition, the situation of the (211) peak of HAp (peak with the highest (100%) intensity) was shifted to the lower angles. The HAp phase was crystallized through a phase transformation of the glass phase to ceramics; the rate constant of this transformation for un-doped, 1, 2.5, 5, and 7.5 mol% Ag-doped MBGs were 7.4 × 10^−7^, 14 × 10^−7^, 13 × 10^−7^, 11 × 10^−7^, and 10 × 10^−7^ s^−1^. 

### 3.5. FTIR Spectroscopy

Figure 5A shows the FTIR spectra of the synthesized MBGs. The specified bands in the range of 570–700 cm^−1^ are related to the symmetrical bending vibration of B-O-B in the [BO_3_] triangles of the pentaborate (B_5_O_8_^−^) groups [26]. The broad bands in the range of (800–1200) cm^−1^ belong to the stretching vibration of [BO_4_] tetrahedra that are associated with di-(B_4_O_7_^2−^), tri-((B_3_O_5_^−^), tetra-(B_8_O_13_^2−^), and pentaborate ((B_5_O_8_^−^) groups [26]. The broad bands in the range of 1200–1600 cm^−1^ are associated with the stretching vibration of the [BO_3_] triangle, which is related to orthoborate groups such as poly-borate [B_3_O_9_]^9−^ and planar [BO_3_]^3−^ [26]. A slight shift to a higher wavenumber was observed for the Ag-doped MBGs compared to their Ag-free counterparts. The FTIR spectra of the SBF-immersed MBGs after 7 days are shown in Figure 5B. The marked bands (553–663) cm^−1^, and (1000–1100) cm^−1^ are related to P-O bands [27]. The observed broad bands in the range of (1460–1560) cm^−1^ are connected with carbonated groups of the hydroxycarbonate apatite (HCA) layer [27]. 

### 3.6. FESEM Observations

The surface morphology of the Ag-free and Ag-doped borate MBGs is presented in Figure 6. As clearly observed, a HAp-like layer was formed on the MBGs during 7 days of incubation in SBF. However, higher percentages (between 5 and 7.5 mol%) of the dopant (Ag) in the glasses can interfere with their bioactivity.

### 3.7. TEM Images

TEM micrographs of the Ag-free and Ag-doped borate-based MBGs are observed in Figure 7. Regarding the glassy nature of samples, the particles do not show specified or oriented morphologies. The porous particles with a 50–100 nm size were observed in the un-doped glass. As observed in the TEM images, doping Ag into the MBGs reduced their particle size to 15–60 nm. It should be pointed out that the well-ordered porosity can be seen in the Ag-doped samples with 7.5 mol%.

### 3.8. AFM Micrographs

The AFM images of MBGs are displayed in Figure 8. As shown, the surface roughness of the MBGs significantly increased (*p* < 0.05) after doping Ag into the samples’ network. The average values of the surface roughness were 35 ± 12, 15 ± 8, 61 ± 6, 6 ± 0.4, and 2 ± 0.2 nm for the un-doped, 1, 2.5, 5, and 7.5 mol% Ag-doped BGs, respectively. It should be noted that the well-dispersed nature of the particles is important for the evaluation of surface roughness. 

### 3.9. N_2_ Adsorption/Desorption Analysis

Figure 9 displays the BET/BJH outcomes of the borate MBGs. Given the results and based on Brunauer–Deming–Deming–Teller theory (BDDT) classification, the glass samples belong to category IV and are closely attributed to the H4 class of mesoporous particles according to [27]. This class is related to irregular and broad-sized and/or ordered mesoporous particles. 

Table 5 represents the mesoporous characteristics of the MBG particles in detail. The S_BET_ of the un-doped, 1, 2.5, 5, and 7.5 mol% Ag-doped MBGs was 47, 87, 104, 145, and 167 m^2^/g, respectively. The corresponding pore radius range of the MBGs was 27–33, 17–25, 29–33, 23–31, and 6–12 nm for the un-doped, 1, 2.5, 5, and 7.5 mol% Ag-doped samples, respectively. The intersection of two adsorption and desorption curves as the representative of adsorption energy for the mentioned MBGs were 0.79, 0.80, 0.74, 0.82, and 0.4, respectively.

### 3.10. Ions Release Profile

The results of the ions released from the borate glasses into SBF are shown in Figure 10A–G. The calculated kinetics of ions released into the SBF are represented in Appendix A. According to the data, B ions were first released from the MBGs into SBF during the first 72 h of incubation, and then they were desorbed to the samples from SBF after 168 h. In the case of P, a constant decrease in its release into SBF was observed over the incubation time (up to 168 h). In contrast, the release of Ca^2+^, Na^+^, and Mg^2+^ ions showed a constant increase during the test. More importantly, the release profile of Ag^+^ ions into SBF indicated a continuous increase during the test period; a burst release can be found in the first 24 h, followed by a slow upward slope in the release by 168 h post-incubation.

### 3.11. pH Variations

The pH changes in the glass containing SBF are shown in Figure 10H. Given the data, the pH was increased from 7.42 to 8.35, 8.42, 8.2, 8.2, and 8.25 for the un-doped, 1, 2.5, 5, and 7.5 mol% Ag-doped glasses.

### 3.12. Cell Viability

The effects of the prepared MBGs on cell viability are shown in Figure 11. According to the data, the viability of cells cultured with the conditioned media containing 5 mg/mL and 3 mg/mL of the MBG particles showed a reduction of up to 20 and 10% compared to control (cells culture without the MBGs), respectively. Incubation of the cells with the conditioned media containing 1 mg/mL of the MBGs had no significant negative effect (*p* > 0.05) on the viability of NIH-3T3 cells during 24 h of incubation. 

### 3.13. Antibacterial Activity

The results of the antibacterial activity of BGs against *P.*
*aeruginosa* and *E.*
*coli* (gram-negative species) as well as *S.*
*aureus* and *B.*
*cereus* (gram-positive species) are shown in Figure 12A,B. According to the data, the un-doped BGs showed 50% antibacterial activity against both gram-positive and negative bacteria. Doping of Ag into BGs significantly (*p* < 0.05) increased the antibacterial activity of BGs by up to 98% against gram-negative bacteria (Figure 12A). Besides, Ag-doping increased the antibacterial activity of BGs by up to 80% against gram-positive bacteria (Figure 12B).

## 4. Discussion

A series of Ag-doped borate 1393B3 MBGs were synthesized through a modified sol–gel process. In this study B(OH)_3_ was used as the precursors of boron. The decomposition of B(OH)_3_ in the water occurred as follows (R1):(R1)$$B(\mathrm{OH}{)}_{3}+2H_{2}O\;\to B(\mathrm{OH}{)}_{4}{}^{-}+H_{3}O^{+}$$

However, the decomposed ions are not stable, and the reaction could easily happen in the opposite direction of the sol–gel process steps (e.g., polycondensation); therefore, a sharp decrease occurs in the homogeneity of the products. Besides, the B(OH)_3_ and another borate source (e.g., tributyl borate) could deposit after evaporation of water from the solutions by heating the sols, as reported elsewhere [28]. To address this problem, along with reducing the costs needed for methoxyethoxide precursors, we used PVA substrate and proceeded with the synthesis at a high pH (pH = 14). In an aqueous media, the B(OH)_3_ and PVA react as follows (R2 and Appendix A):(R2)$$B(\mathrm{OH}{)}_{3}+2C_{2}H_{4}O+2H_{2}O\;\to \;(C_{2}H_{4}O{)}_{2}*B(OH{)}_{4}{}^{-}+H_{3}O^{+}$$
where C_2_H_4_O is a single unit of PVA structure, and (C_2_H_4_O)_2_*B(OH)_4_^−^ is a single unit of polyvinyl borate ion’s structure.

This reaction impedes the B(OH)_3_ depositions in all the sol–gel processes. Moreover, increasing the pH of PVA-containing solutions could lead to in-situ depositions of all the used nitrate compounds. The results of the XRD pattern of the formed gels after this in-situ deposition indicated the presence of PVA in the structures (Appendix A). In the heat-treatment process, the oxides or multi-component BGs could be formed. According to the following reaction (R3), the nucleation of the oxides instead of the BGs in low temperatures is very insignificant.
(R3)$$54.6B_{2}O_{3}+14.6\mathrm{CaO}+7.5{\mathrm{Ag}}_{2}O+7.6\mathrm{MgO}+7.9K_{2}O+6.1{\mathrm{Na}}_{2}O+1.7P_{2}O_{5}\;\to (B_{2}O_{3}{)}_{54.6}(\mathrm{CaO}{)}_{14.6}({\mathrm{Ag}}_{2}O{)}_{7.5}(\mathrm{MgO}{)}_{7.6}(K_{2}O{)}_{7.9}({\mathrm{Na}}_{2}O{)}_{6.1}(P_{2}O_{5}{)}_{1.7}$$

The calculated equilibrium constant (K) of this reaction is about 10^308^, which indicates the formation of multi-component oxides instead of single ones. The minor amounts of the unwanted calcium borate phase were nucleated in our experimental section. Although this phase is water-soluble and could easily be removed from the BGs, the optimization of the process needs more investigation in future studies to avoid the formation of any unwanted phase.

The glassy state of the particles was increased by increasing the amounts of dopants (Table 4). From a thermodynamic point of view, the stability of multi-component BGs is higher (the higher value of entropy or higher structural disorder) [29]. Our calculations and estimations with HSC chemistry^®^ software showed that the entropy of the un-doped MBGs is 5818.138 J/K, and doping Ag up to 7.5 mol% increased the value to 6506.882 J/K. Besides, the calculated values of entropy for the un-doped and the 7.5 mol% Ag-doped samples are 38.048 R and 39.75 R, respectively (R is the gas constant and equals 8.314 J/mol × K). According to these calculations, the disordering of BGs in multi-components or the doped BGs is increased and prevents their nucleation in the sol–gel process.

Basically, Ag-doping of the glass network could break the network’s bonds and generate non-bridging oxygen (NBO) [29]. As reported, the Urbach energy of the glass network is enhanced by increasing the amount of the Ag_2_O modifier [30]. This increment in energy could raise the disordering of the glass network [30] and thereby lead to changes in the thermal behavior of the glass. According to the DTA results (Figure 2), T_g_ decreased in the Ag-doped MBGs from 573 to 526 °C. The results confirmed the effect of Ag-doping on preventing particle growth (the particle size was reduced from 105 to 25 nm) (Table 3). As reported [31,32], the creation of NBO in the glass network could increase the mobility of ions; thus, particle growth is impeded. The effect of Ag_2_O doping on the borate glass network is presented as follows (R4):(R4)[>−O−OB−O−B <O−O−]+Ag2O→[>−O−OB−O−]+2Ag+

The zeta potential and mobility measurements showed an increase in the surface charge and mobility of the Ag-doped samples. The zeta potential and mobility of the Ag-free MBGs were −12 mV and −0.89 ± 0.2, which increased to −22 mV and −1.58 ± 0.3 μ/s/V/cm in the case of the MBG-Ag7.5), respectively. 

A broad hump was observed in the XRD patterns of the MBGs (Figure 4), indicating the amorphous nature of the synthesized glasses. The crystalline phase appeared in the XRD in the case of SBF-doped glasses. The Ag-doping could increase the rate of glass to HAp transformation in all the samples and result in an increase in the rate constant of transformation from 7.4 × 10^−7^ up to 14 × 10^−7^ s^−1^. As our calculation shows, the peak position and the lattice constant of HAp were changed while soaking the MBGs in SBF. This may be associated with the doping of the released ions (such as B and Ag^+^) from SBF to the HAp network. The FTIR study (Figure 5A) confirmed the existence of borate groups in all the samples. The obtained data is clear evidence for the structural modifications and the creation of NBOs in the structure of Ag-doped MBGs. The increased concentration of NBOs could lead to an enhancement in the ionic conductivity of the MBGs. Furthermore, the structural bands of HAp were detected after immersion of all the MBGs in SBF (Figure 5B).

The surface morphology of the MBGs before and after immersion in SBF (Figure 6) displayed the formation of the HCA layer onto the MBGs, which is in line with the XRD and FTIR results. The TEM micrographs (Figure 7) of the prepared glasses revealed that the presence of the dopant (Ag) up to 7.5 mol% decreased the particle size from 50 to 100 nm to below 15–60 nm. Moreover, the surface roughness of the MBGs was investigated using AFM analysis (Figure 8), and the obtained data uncovered the significant impact of the dopant on the MBGs surface. Recently, the gradient distribution of the dopants from the bulk to the surface of BGs was reported [33]. According to the AFM data, the surface roughness of the BGs was changed from 35 (Ag0) to 2 nm (Ag7.5). Furthermore, the dopants could act as the functionalization agent and generate surface defects (e.g., oxygen vacancies). Increasing the zeta potential values of the doped samples (Table 3) was in line with this hypothesis.

A summary of the mesoporous characteristics is given in Table 5. According to the data, all the glass samples have a mesoporous nature (pore size between 2 and 50 nm). The mesoporous structure of the samples is related to the nature of the sol–gel process as well as the burning out of the PVA template during the calcination process [34]. Increasing the amounts of S_BET_ of the Ag-doped (from 47 (Ag0) to 167 m^2^/g (Ag7.5) samples (Figure 9) is correlated to changing the morphology and integrity of the particles as well as changing the ionic conductivity, as reported elsewhere [35,36]. The ion’s release profile indicated a burst release for all the ions (except P^5+^) during the first hours of incubation in SBF (Figure 10). The study of the kinetics of ions released in SBF (Appendix A) demonstrated the independence of the ions released from the amount of the dopant (Ag).

The MTT assay (Figure 11) showed the glasses have no negative effects on the viability of NIH3T3 fibroblast cells at a concentration of 1 mg/mL. However, a significant inhibitory effect was detected in concentrations higher than 3 mg/mL of the MBGs. BGs were previously reported to have inhibitory impacts on cells’ proliferation due to the burst release of ions from their network to the surrounding biological environment and also a sudden increase in pH. All these events can be moderated in vivo conditions due to the large volume of body fluids [37]. It has been reported that 1393B3 glasses have an intrinsic activity against bacterial growth and proliferation [38], due to the high release of ions from their structure into the environment. Our study demonstrated the antibacterial activity of the Ag-free MBGs (50% against both gram-positive and gram-negative bacteria), which was significantly increased after doping with Ag^+^ ions (up to 80 and 98% against gram-positive and gram-negative species, respectively) (Figure 12). Previously, it has been clarified that Ag^+^ ions can (I) disrupt the bacterial cell wall and cytoplasmic membrane; (II) facilitate the denaturation of ribosomes; (III) interrupt the production of adenosine triphosphate (ATP); (IV) elevate ROS production; (V) interfere with the DNA replication [39]. Interestingly, silver was reported to have more lethal effects on gram-negative than gram-positive bacteria due to their different metabolism profile [20,40].

## 5. Conclusions

The Ag-doped 1393B3 borate MBGs were successfully synthesized through a modified sol–gel process using nitrate precursors. We used PVA substrate and implemented pH = 14 for stabilizing boron in the solution and deposition of all the nitrate precursors, respectively. The XRD results revealed the amorphous state of the Ag-free and Ag-doped MBGs. The particle size, zeta potential, surface roughness, and S_BET_ values were 105 nm, −12 mV, 35 nm, and 47 m^2^/g for the Ag-free borate MBGs and changed to 25 nm, −22 mV, 2 nm, and 167 m^2^/g for MBG-Ag7.5. The incorporation of Ag at concentrations of 1, 2.5, and 5 (mol%) into the glass composition had no adverse effect on the bioactivity; however, 7.5 mol% of this dopant hinders the surface reactivity of the glasses. The FTIR spectroscopic study confirmed that doping Ag to the MBGs may increase the NBO concentrations. From a biological point of view, Ag-doped borate MBGs had no inhibitory effects on the growth and proliferation of fibroblast cells at a concentration of 1 mg/mL. More importantly, adding Ag to the borate BGs improved their antibacterial activity from 50% to about 80% and 98% for gram-positive and gram-negative bacteria. In summary, our study showed that Ag-doped 1393B3 MBGs are suitable antibacterial substances and can be utilized for a wide range of wound healing applications, such as the management of burns.

## Figures and Tables

**Figure 1 bioengineering-09-00442-f001:**
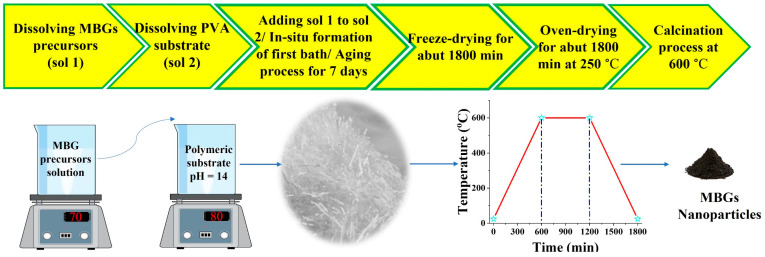
Schematic representation of the sol–gel synthesis process of 1393B3 borate BGs by using a polymeric substrate (PVA).

**Figure 2 bioengineering-09-00442-f002:**
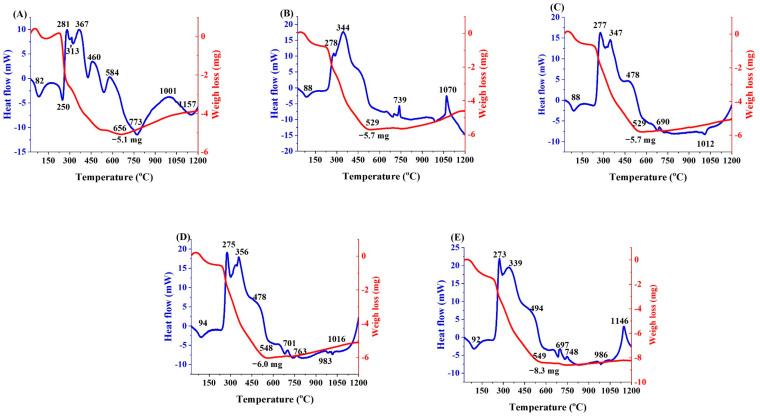
(**A**–**E**): DTA/TGA graphs of the Ag-free (**A**) and the Ag-doped (1, 2.5, 5, 7.5 mol%) MBGs (**B**–**E**).

**Figure 3 bioengineering-09-00442-f003:**
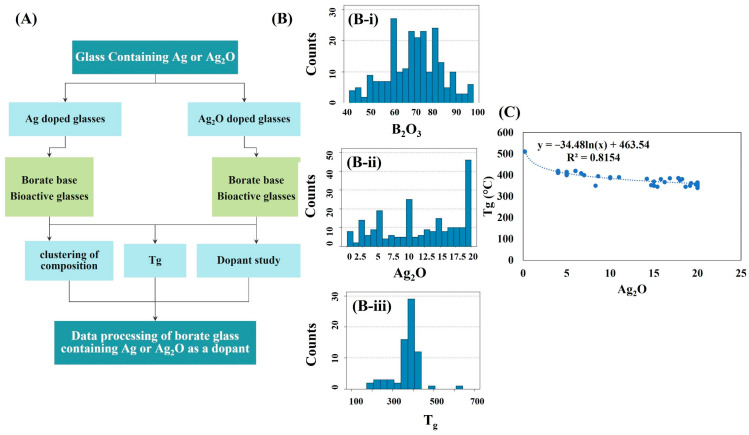
(**A**–**C**) The adapted data from the Sci-glass database, including decision tree (**A**), frequency distribution histogram of B_2_O_3_ (**B-i**), Ag_2_O (**B-ii**), and T_g_ (**B-iii**) (**B**), and the correlation of T_g_ (°C) (y) with the concentration of Ag_2_O (mol%) (**C**).

**Figure 4 bioengineering-09-00442-f004:**
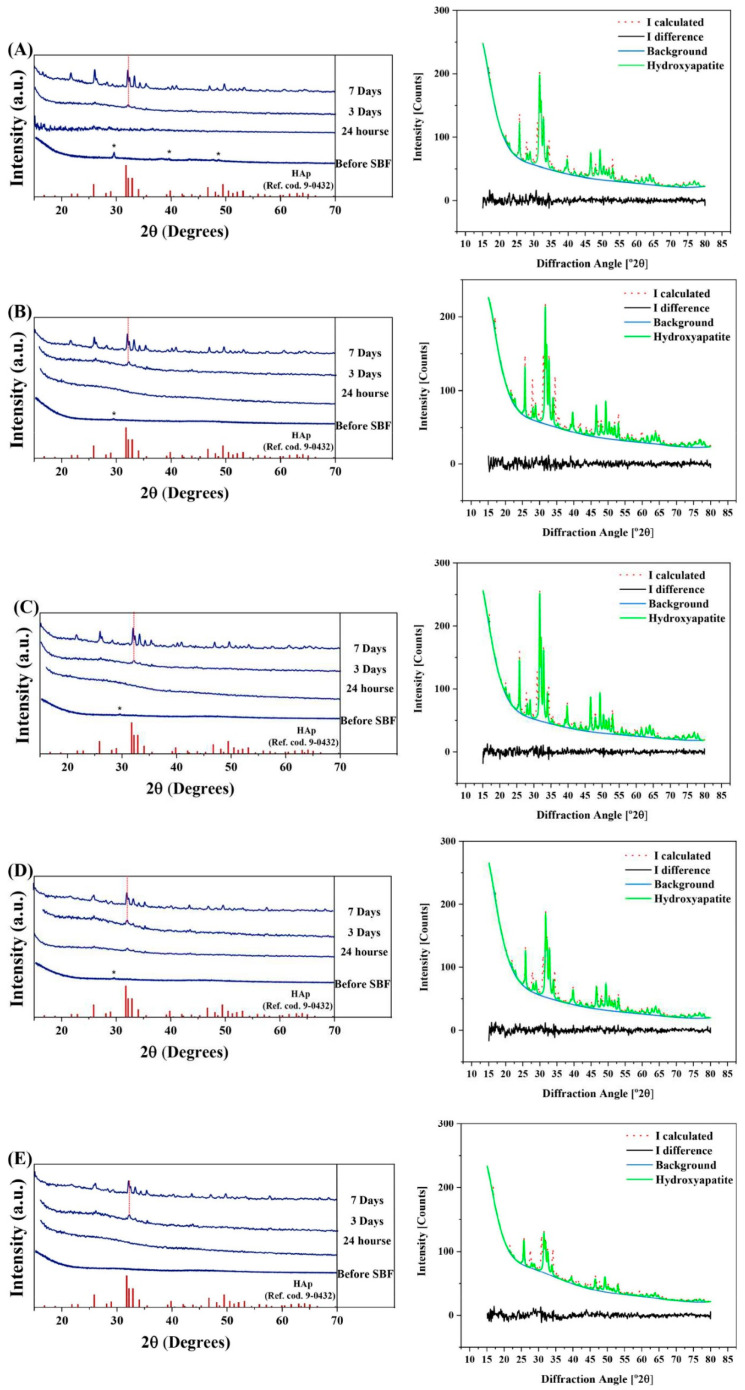
XRD patterns and Rietveld refinement study of the un-doped (**A**), 1 (**B**), 2.5 (**C**), 5 (**D**), and 7.5 (**E**) mol% Ag-doped MBGs. The observed peaks (labeled with *) in the XRD patterns of the samples before soaking in SBF are related to calcium borate (CaB_2_O_4_, ICDD ref. cod. 01-076-0747, orthorhombic).

**Figure 5 bioengineering-09-00442-f005:**
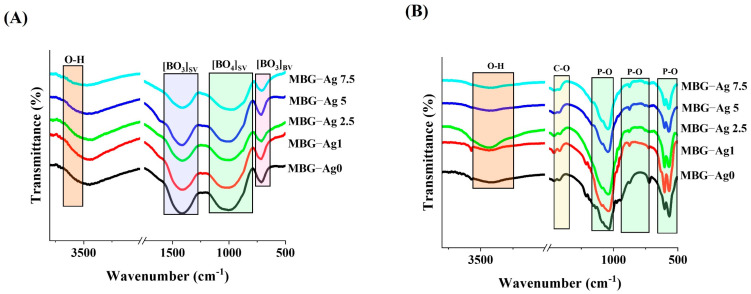
FTIR spectra of the Ag-free and Ag-doped MBGs before (**A**) and after immersion in SBF (**B**).

**Figure 6 bioengineering-09-00442-f006:**
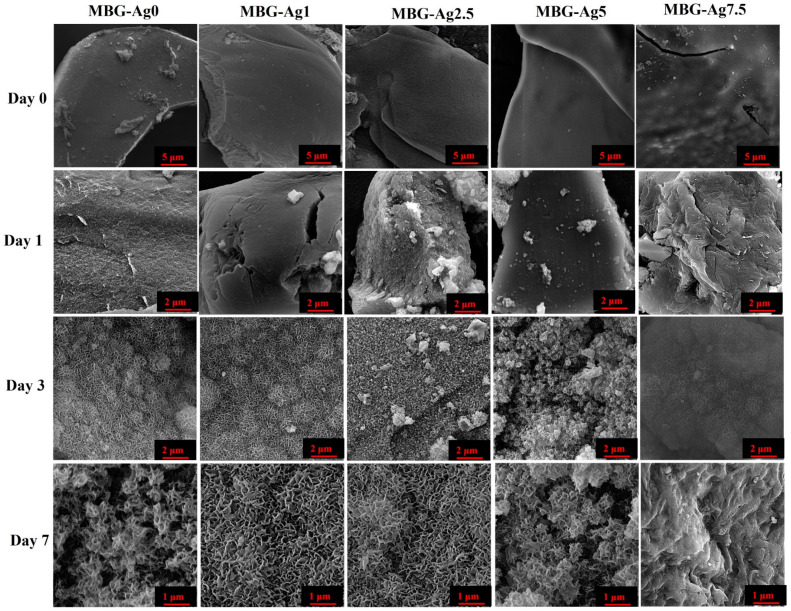
FESEM images of un-doped and Ag-doped MBGs before and after immersion in SBF.

**Figure 7 bioengineering-09-00442-f007:**
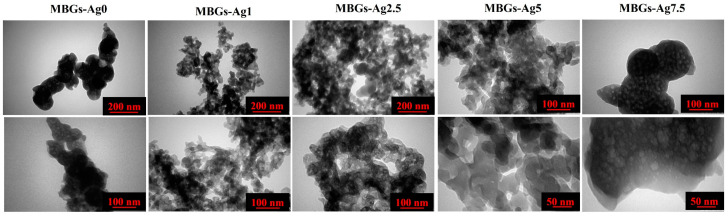
TEM images of the Ag-free and Ag-doped MBGs.

**Figure 8 bioengineering-09-00442-f008:**
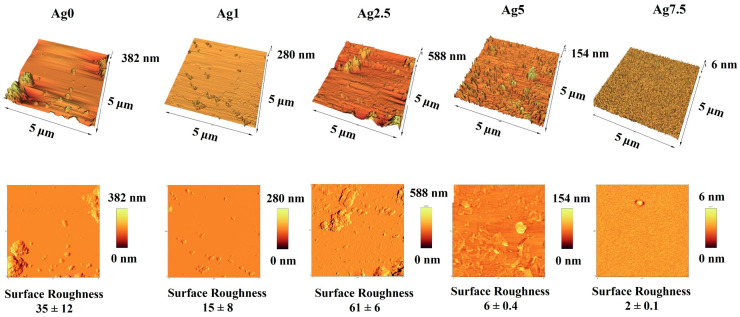
AFM images of the Ag-free and 1, 2.5, 5, 7.5 mol% Ag-doped 1393B3 MBGs.

**Figure 9 bioengineering-09-00442-f009:**
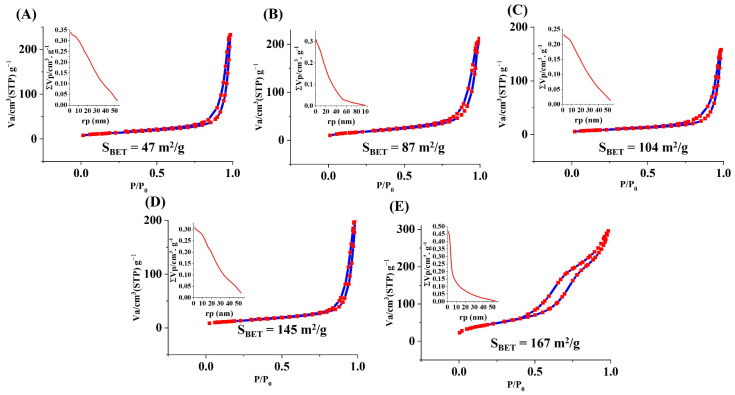
N_2_ adsorption/desorption graphs of (**A**) the un-doped and (**B**–**E**) 1, 2.5, 5, and 7.5 mol% Ag-doped MBGs.

**Figure 10 bioengineering-09-00442-f010:**
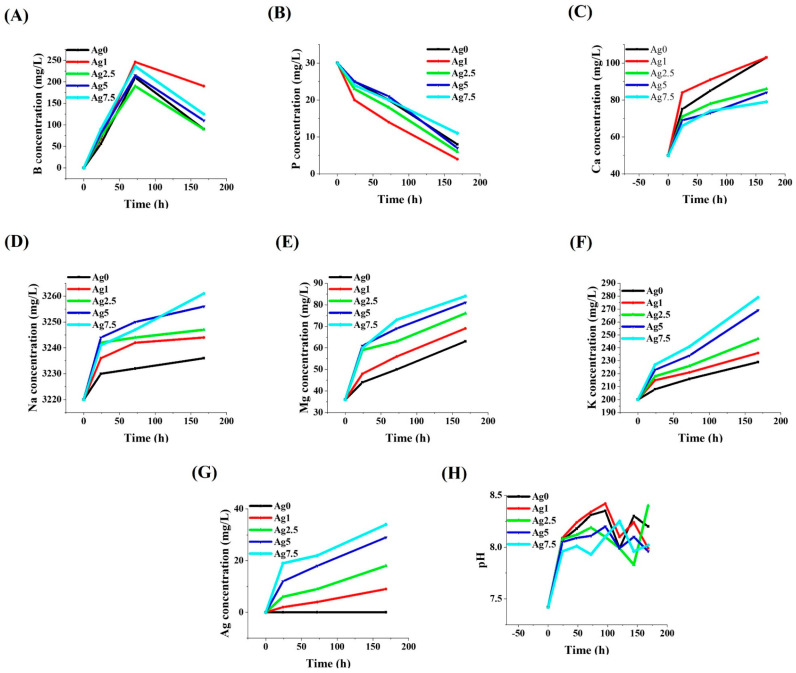
The release profile of different ions from Ag-free and Ag-doped MBGs (**A**–**G**), and the pH changes of the MBGs-containing SBF (**H**).

**Figure 11 bioengineering-09-00442-f011:**
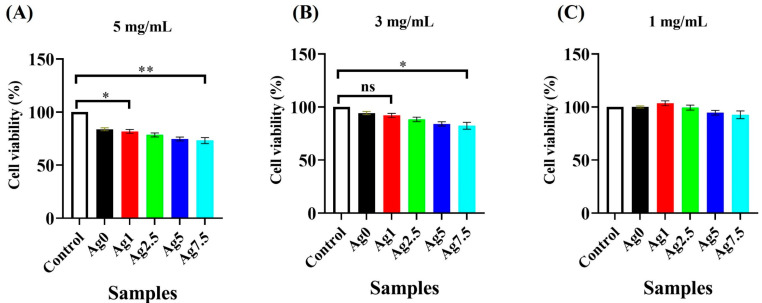
MTT assay results of the Ag-free and Ag-doped MBGs. (* *p* ≤ 0.05, ** *p* ≤ 0.01, ns means not significant).

**Figure 12 bioengineering-09-00442-f012:**
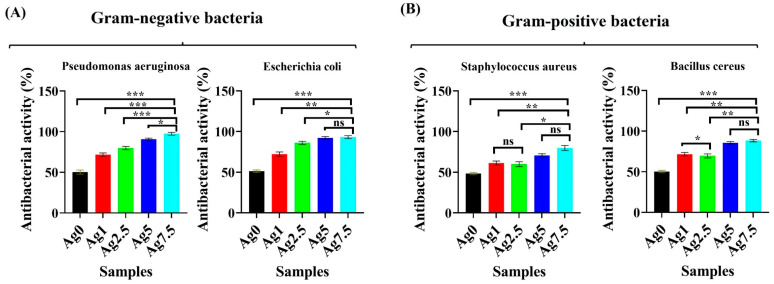
The result of antibacterial activity of the un-doped and Ag-doped 1393B3 MBGs against gram-negative (**A**) and gram-positive (**B**) bacteria species. (* *p* ≤ 0.05, ** *p* ≤ 0.01, *** *p* ≤ 0.001, ns means not significant).

**Table 1 bioengineering-09-00442-t001:** Nominal compositions of the Ag-doped 1393B3 MBGs (mol.%).

Sample	B_2_O_3_	CaO	Ag_2_O	MgO	K_2_O	Na_2_O	P_2_O_5_
MBG-Ag0	54.6	22.1	0	7.6	7.9	6.1	1.7
MBG-Ag1	54.6	21.1	1	7.6	7.9	6.1	1.7
MBG-Ag2.5	54.6	19.6	2.5	7.6	7.9	6.1	1.7
MBG-Ag5	54.6	17.1	5	7.6	7.9	6.1	1.7
MBG-Ag7.5	54.6	14.6	7.5	7.6	7.9	6.1	1.7

**Table 2 bioengineering-09-00442-t002:** Ion release concentrations of the synthesized Ag-free and Ag-doped 1393B3 MBGs.

Sample		B_2_O_3_	CaO	Ag_2_O	MgO	K_2_O	Na_2_O	P_2_O_5_
MBG-Ag0	ICP (mol%)	54.74 ± 0.20	22.08 ± 0.05	-	7.58 ± 0.30	7.85 ± 0.10	6.10 ± 0.05	1.62 ± 0.21
	Designed (mol%)	54.6	22.1	-	7.6	7.9	6.1	1.7
MBG-Ag1	ICP (mol%)	54.6 ± 0.21	21.12 ± 0.05	0.90 ± 0.3	7.60 ± 0.02	7.91 ± 0.02	6.1 ± 0.05	1.7 ± 0.23
	Designed (mol%)	54.6	21.1	1	7.6	7.9	6.1	1.7
MBG-Ag2.5	ICP (mol%)	54.7 ± 0.2	19.64 ± 0.06	2.3 ± 0.05	7.62 ± 0.03	7.92 ± 0.04	6.1 ± 0.01	1.7 ± 0.04
	Designed (mol %)	54.6	19.6	2.5	7.6	7.9	6.1	1.7
MBG-Ag5	ICP (mol%)	54.7 ± 0.03	17.03 ± 0.10	4.92 ± 0.01	7.61 ± 0.03	7.91 ± 0.05	6.11 ± 0.04	1.71 ± 0.05
	Designed (mol%)	54.6	17.1	5	7.6	7.9	6.1	1.7
MBG-Ag7.5	ICP (mol%)	54.74 ± 0.06	14.90 ± 0.20	7.01 ± 0.02	7.60 ± 0.03	7.90 ± 0.03	6.1 ± 0.04	1.72 ± 0.03
	Designed (mol%)	54.6	14.6	7.5	7.6	7.9	6.1	1.7

**Table 3 bioengineering-09-00442-t003:** DLS and surface charge characteristics of MBGs. (** *p* ≤ 0.01, *** *p* ≤ 0.001, **** *p* ≤ 0.0001).

	DLS Characteristics		Surface Charge Characteristics				
Sample	Dv50 (nm)	Dv90 (nm)	*p* Value	Zeta (mV)	*p* Value	Mobility μ/s/V/cm	*p* Value
MBG-Ag0	51 ± 3	105 ± 8	-	−12 ± 2	-	−0.89 ± 0.2	-
MBG-Ag1	17 ± 2	50 ± 10	***	−16 ± 2	**	−1.07 ± 0.1	***
MBG-Ag2.5	12 ± 1	40 ± 6	***	−18 ± 3	**	−1.18 ± 0.1	***
MBG-Ag5	8 ± 4	25 ± 5	****	−20 ± 2	**	−1.28 ± 0.1	***
MBG-Ag7.5	44 ± 6	54 ± 9	****	−22 ± 1	***	−1.58 ± 0.3	***

**Table 4 bioengineering-09-00442-t004:** Rietveld refinement study of the un-doped, 1, 2.5, 5, and 7.5 mol% Ag-doped MBGs.

Sample	Crystallite Size (nm)		Crystallinity (%)				a Å		c Å		Average Rate Constant (K × 10^−7^) (s^−1^)
Day	3	7	0	1	3	7	3	7	3	7	-
**MBG-Ag0**	8	38	11	8	12	35	9.438	9.442	6.864	6.895	7.4
**MBG-Ag1**	4	43	7	4	16	41	9.429	9.433	6.893	6.893	14
**MBG-Ag2.5**	9	41	4	4	12	39	9.431	9.433	6.896	6.898	13
**MBG-Ag5**	7	35	4	6	15	36	9.438	9.440	8.899	6.902	11
**MBG-Ag7.5**	4	31	<1	3	6	20	9.442	9.449	6.890	6.891	10

**Table 5 bioengineering-09-00442-t005:** The mesoporous characteristics of the Ag-free and 1, 2.5, 5, and 7.5% Ag-doped MBGs.

Sample	S_BET_ (m^2^/g)	Total Pore Volume (cm^3^/g)	Range of Pore Diameter (nm)	P/P_0_
MBG-Ag0	47	0.36	27–33	0.79
MBG-Ag1	87	0.31	17–25	0.80
MBG-Ag2.5	104	0.24	29–33	0.74
MBG-Ag5	145	0.31	23–31	0.82
MBG-Ag7.5	167	0.45	6–12	0.40

## Data Availability

Data will be made available on request.

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
