# Peer review of "Modified Sol–Gel Synthesis of Mesoporous Borate Bioactive Glasses for Potential Use in Wound Healing"

_bioengineering, 2022, doi:10.3390/bioengineering9090442_

Round 1
Reviewer 1 Report
This article illustrates the potential of using mesoporous borate bioactive glasses in wound healing. The author comprehensively characterized the prepared materials and compared the compound effects. However, some characterization should be explained further. On the other hand, the author identified that Ag-doped mesoporous bioactive glass (MBGs) exhibited potent antibacterial activity against gram-positive and gram-negative bacteria, while more explanation on the selective might be helpful. Below are some comments for your consideration.
1. To avoid misinterpretation, the technical abbreviations in the Materials and Methods section should be referred to in full. For example, in line 123, the “XRD” is in short form, while in line 173, it is labelled as “X-ray diffraction”. It would be better if the names of specific techniques were used as subtitles in 2.2. Similar corrections should be made throughout the whole manuscript.
2. Are ICP results in section 3.1 repeated three times? The deviation seems pretty good for the high (B2O3) and low (P2O5) concentrations. Is that normal?
3. The images in figure 3 are too small, and it is better to increase the whole size of figure 3. Here, the R2=0.8154; could it reach 0.95 by repeated experiments with better parameters set up?
4. Figure 4 is the same case as figure 3 and needs to increase the quality, especially the number inside the figures. Why does doping Ag or Ag2O to borate-based compounds reduce their Tg? Could you explain more here?
5. Sections 3.3 DLS characterization and section 3.4 Zeta potential analysis might be combined into one section as both use a DLS machine to do characterizations. Or change the “3.3 DLS characterization” to “3.3 Size characterization”.
6. In line 266, the authors claimed, “The XRD patterns of the BGs before immersion in SBF confirmed a major amorphous state of the synthesized particles with traces of crystalline calcium borate”, but it did not mention which figure or tip they drew this conclusion from.
7. Figure 8 is the same case as figure 3 and needs to increase the quality. Compared with Ag5 and Ag1, it seems that Ag 5 is much rougher. And the particle is not evenly distributed; is the roughness from AFM meaningful?
8. Any selectively of the materials on the gram-positive and gram-negative bacteria? Could you provide more explanation on the mechanism and selectivity?
9. Please check the typos in the manuscript. I did not find labels A and B in figure 5. In line 152, why use the bold font?
Author Response
Reviewer #1:
******************************************************************************
Comments and Suggestions for Authors
This article illustrates the potential of using mesoporous borate bioactive glasses in wound healing. The author comprehensively characterized the prepared materials and compared the compound effects. However, some characterization should be explained further. On the other hand, the author identified that Ag-doped mesoporous bioactive glass (MBGs) exhibited potent antibacterial activity against gram-positive and gram-negative bacteria, while more explanation on the selective might be helpful. Below are some comments for your consideration.
Authors’ response (AR): We are thankful to Reviewer for taking his/her time for reading and put comments on the current research. In the revised version, we tried to improve the work’s quality and address all Reviewer’ comments.
- To avoid misinterpretation, the technical abbreviations in the Materials and Methods section should be referred to in full. For example, in line 123, the “XRD” is in short form, while in line 173, it is labelled as “X-ray diffraction”. It would be better if the names of specific techniques were used as subtitles in 2.2. Similar corrections should be made throughout the whole manuscript.
AR: All abbreviations were modified throughout the manuscript.
- Are ICP results in section 3.1 repeated three times? The deviation seems pretty good for the high (B2O3) and low (P2O5) concentrations. Is that normal?
AR: The ICP data was checked again. It is OK.
- The images in figure 3 are too small, and it is better to increase the whole size of figure 3. Here, the R2=0.8154; could it reach 0.95 by repeated experiments with better parameters set up?
AR: The quality of the mentioned image was improved. The presented data in figure 3 was adapted from the SciGlass database and after data cleaning, the data with the maximum possible R2 was presented.
- Figure 4 is the same case as figure 3 and needs to increase the quality, especially the number inside the figures. Why does doping Ag or Ag2O to borate-based compounds reduce their Tg? Could you explain more here?
AR: The quality of the image was improved. Some sentences were added to the discussion to cover your invaluable point about Tg.
- Sections 3.3 DLS characterization and section 3.4 Zeta potential analysis might be combined into one section as both use a DLS machine to do characterizations. Or change the “3.3 DLS characterization” to “3.3 Size characterization”.
AR: The sections were combined.
- Figure 8 is the same case as figure 3 and needs to increase the quality. Compared with Ag5 and Ag1, it seems that Ag 5 is much rougher. And the particle is not evenly distributed; is the roughness from AFM meaningful?
AR: The quality of the figure was improved. Reviewer is right, we added some sentences to the text to cover the reviewer's invaluable point.
- In line 266, the authors claimed, “The XRD patterns of the BGs before immersion in SBF confirmed a major amorphous state of the synthesized particles with traces of crystalline calcium borate”, but it did not mention which figure or tip they drew this conclusion from.
AR: The relative data was added to the text.
- Any selectively of the materials on the gram-positive and gram-negative bacteria? Could you provide more explanation on the mechanism and selectivity?
AR: Thanks for mentioning this important point. Previous experiments have shown that Ag is more effective on Gram-negative bacteria than on Gram-positive bacteria (Appl Environ Microbiol. 2008 Apr;74(7):2171-8. doi: 10.1128/AEM.02001-07 and J. Phys. Chem. C 2011, 115, 13, 5461–5468) due to their different metabolism profile, and our results are in consistence with them.
- Please check the typos in the manuscript. I did not find labels A and B in figure 5. In line 152, why use the bold font?
AR: The text was mended for eliminating unwanted typos.

Reviewer 2 Report
This paper seems to have originality a little. But introduction and data presentation should be thoroughly revised.
(1) Introduction should be improved. There are several reports on borate-based bioactive glass for wound healing. However, the reviewer could not well understand what is improved in the present study.
Wray P. Cotton candy’ that heals? Borate glass nanofiber look promising. Am Ceram Bull. 2011;90:25–29.
Yamaguchi, S., Takeuchi, T., Ito, M. et al. CaO-B2O3-SiO2 glass fibers for wound healing. J Mater Sci: Mater Med 33, 15 (2022).
(2) Figure 4 is too small to see. Layout should be improved. Also, what kind of crystalline phase is formed before soaking (several marked peaks)? This should be clarified in the figure.
(3) In Fig 5B, assignment is not correct in some parts. Namely, the region in 1000-1100 cm-1 is marked as C-O.
Author Response
Reviewer #2:
******************************************************************************
This paper seems to have originality a little. But introduction and data presentation should be thoroughly revised.
Authors’ response (AR): We are thankful to Reviewer for taking his/her time for reading and comment on the current research. In the revised version, we tried to improve the work’s quality and address all Reviewer’ comments.
(1) Introduction should be improved. There are several reports on borate-based bioactive glass for wound healing. However, the reviewer could not well understand what is improved in the present study.
Wray P. Cotton candy’ that heals? Borate glass nanofiber look promising. Am Ceram Bull. 2011;90:25–29.
Yamaguchi, S., Takeuchi, T., Ito, M. et al. CaO-B2O3-SiO2 glass fibers for wound healing. J Mater Sci: Mater Med 33, 15 (2022).
AR: As reviewer mentioned, there are several studies on the effectiveness of borate BGs on wound healing. However, our study is the first report on the successful preparation of sol-gel borate BGs using nitrate precursors instead of expensive counterparts (i.e., methoxy precursors). This may be interesting for the community for preparing such glasses inexpensively. All in all, we improved the introduction according to the mentioned papers.
(2) Figure 4 is too small to see. Layout should be improved. Also, what kind of crystalline phase is formed before soaking (several marked peaks)? This should be clarified in the figure.
AR: The quality of the mentioned image was improved.
(3) In Fig 5B, assignment is not correct in some parts. Namely, the region in 1000-1100 cm-1 is marked as C-O.
AR: The modifications were made to address your invaluable comment.

Round 2
Reviewer 1 Report
The authors addressed all of my concerns. Thanks.
Reviewer 2 Report
This is well revised according to the reviewer's comments.